# The Various Packing Structures of Tb@C_82_ (I, II) Isomers in Their Cocrystals with Ni(OEP)

**DOI:** 10.3390/nano13060994

**Published:** 2023-03-09

**Authors:** Wei Dong, Qin Zhou, Wangqiang Shen, Le Yang, Peng Jin, Xing Lu, Yongfu Lian

**Affiliations:** 1Key Laboratory of Functional Inorganic Material Chemistry, Ministry of Education, School of Chemistry and Materials Science, Heilongjiang University, Harbin 150080, China; 2State Key Laboratory of Materials Processing and Die & Mould Technology, School of Materials Science and Engineering, Huazhong University of Science and Technology, 1037 Luoyu Road, Wuhan 430074, China; 3School of Materials Science and Engineering, Hebei University of Technology, Tianjin 300130, China

**Keywords:** metallofullerene, terbium, cocrystals, centrosymmetric dimer, unpaired spin

## Abstract

Soot-containing terbium (Tb)-embedded fullerenes were prepared by evaporation of Tb_4_O_7_-doped graphite rods in an electric arc discharge chamber. After 1,2,4-trichlorobenzene extraction of the soot and rotary evaporation of the extract, a solid product was obtained and then dissolved into toluene by ultrasonication. Through a three-stage high-pressure liquid chromatographic (HPLC) process, Tb@C_82_ (I, II) isomers were isolated from the toluene solution of fullerenes and metallofullerenes. With the success of the growth of cocrystals of Tb@C_82_ (I, II) with Ni(OEP), the molecular structures of Tb@C_82_ (I) and Tb@C_82_ (II) were confirmed to be Tb@C*_2v_*(9)-C_82_ and Tb@C*s*(6)-C_82_, respectively, based on crystallographic data from X-ray single-crystal diffraction. Moreover, it was found that Tb@C_82_ (I, II) isomers demonstrated different packing behaviors in their cocrystals with Ni(OEP). Tb@C*_2v_*(9)-C_82_ forms a 1:1 cocrystal with Ni(OEP), in which Tb@*C_2v_*(9)-C_82_ is aligned diagonally between the Ni(OEP) bilayers to form zigzag chains. In sharp contrast, Tb@C*s*(6)-C_82_ forms a 2:2 cocrystal with Ni(OEP), in which Tb@C*s*(6)-C_82_ forms a centrosymmetric dimer that is aligned linearly with Ni(OEP) pairs to form one-dimensional structures in the a–c lattice plane. In addition, the distance of a Ni atom in Ni(OEP) to the C*s*(6)-C_82_ cage is much shorter than that to the C*_2v_*(9)-C_82_ one, indicative of a stronger π-π interaction between Ni(OEP) and the C_82_ carbon cage in the cocrystal of Tb@C*_S_*(6)-C_82_ and Ni(OEP). Density functional theory calculations reveal that the regionally selective dimerization of Tb@C*_S_*(6)-C_82_ is the result of a dominant unpaired spin existing on a particular C atom of the C*_S_*(6)-C_82_ cage.

## 1. Introduction

Endohedral metallofullerenes (EMFs) are formed by embedding metal atoms or clusters of metal and nonmetal atoms inside the carbon cage of a fullerene molecule. Due to their unique molecular and electronic structures, EMFs are superior to fullerenes for their extensive applications in the fields of medical diagnostic and therapeutic reagents, organic photovoltaic power generation, organic superconductors, and single-molecule magnets [1,2,3,4,5,6,7]. To date, EMFs have been a hotspot for investigations of a series of novel nanocarbon materials.

Among the large number of EMFs, C_82_-based mono-EMFs (M@C_82_, M = encapsulated metal atom) are of particular interest due to their availability, solubility, and redox activity [8,9,10]. Based on the formal charge of the embedded metal atom, mono-EMFs can be classified as divalent group M^2+^@C_82_^2−^ (M = Sm, Eu, Tm, Yb and alkaline earth metals) [11,12,13,14,15], trivalent group M^3+^@C_82_^3−^ (M = Sc, Y, La, Ho, Ce, Pr, Nd, Gd, Dy, Er and Lu) [16,17,18,19,20,21,22,23,24], or tetravalent group M^4+^@C_82_^4−^ (M = Th and U) [25,26]. Elucidation of the molecular and crystallographic structures of C_82_-based mono-EMFs plays a crucial role for understanding the interactions between encapsulated metal atoms and the C_82_ carbon cage, as well as the stability, growth mechanism, and packing behavior of M@C_82_ molecules.

As one of the rare earth elements, terbium is of special interest. On the one hand, terbium has variable valence states. As the starting material for the synthesis of EMFs, Tb_4_O_7_ is a mixture of oxides with two valence states of +3 and +4. Accordingly, study of Tb-containing mono-EMFs would be beneficial to clarify the electronegative nature of fullerene cages as well as the electronic structure of EMFs [27]. On the other hand, terbium is a well-known rare earth luminescent material, and its compounds have better performance in optical properties. Moreover, Tb_2_@C_80_(CH_2_Ph), as a single molecular magnet demonstrates, has a very strong coercivity and the highest blocking temperature (28.9 K) of the di-nuclear lanthanide complexes [28]. Therefore, Tb-containing EMFs might be applicable in many fields as novel optical and magnetic materials.

Investigation of the molecular and crystalline structures of EMFs is fundamental for their potential applications. Previously, some Tb-containing clusterfullerenes, including Tb_3_N@C_2n_(2n = 80, 86 and 88) [29], TbCN@C_82_ [30], TbCN@C_76_ [31], and Tb_2_C_2_@C_82_ [32] have been successfully synthesized and separated, and their molecular and crystalline structures have been determined. It was reported for TbCN@C_82_ and TbCN@C_76_ that TbNC clusters are flexible, and the geometry of the embedded TbNC clusters and the length of the Tb-N(C)/C-N bond can be greatly changed by changing the size of the isomer or carbon cage structure. However, the crystalline structures of Tb-containing mono-EMFs have not been reported yet, even though some of them have been isolated and characterized electrochemically and spectroscopically [33,34].

On the other hand, with the successful preparation of cocrystals of mono-EMF and metal octaethylporphyrin M(OEP), the packing behaviors of some mono-EMFs have been clarified. Apart from the normal packing of individual mono-EMFs, the dimerization of some mono-EMFs, including Li@C_60_ [35], Y@C*s*(6)-C_82_ [36], Er@C*s*(6)-C_82_ [37], and Ce@C*_2v_*(9)-C_82_ [38] occurs in their cocrystals with M(OEP). It was found that the formed dimers are linked by a single C–C interfullerene bond.

In this work, two isomers of Tb@C_82_ were isolated, and their cocrystals with Ni(OEP) were successfully prepared. Crystallographic characterization revealed that the molecular structures of Tb@C_82_ (I, II) could be separately described as Tb@C*_2v_*(9)-C_82_ and Tb@C*_S_*(6)-C_82_, and that the labile isomer demonstrated a high degree of regioselective dimerization in its crystalline state. Theoretical calculations indicate that the varied anisotropy of the cage spin density, determined by carbon cage geometry and the metal center position, is responsible for the different packing behaviors of the two Tb@C_82_ (I, II) isomers.

## 2. Materials and Methods

### 2.1. Synthesis, Isolation and Characterization

Soot-containing Tb@C_82_ isomers were prepared by a modified arc discharge method. The anode was a graphite rod (Ø8 × 140 mm, Sinosteel Shanghai Carbon Plant, Shanghai, China) drilled with a hole (Ø6 × 120 mm) and filled with a powder mixture of Tb_4_O_7_ (Aladdin, Shanghai, China) and graphite (Tianjin Kermel Chemical Reagent Co.,Ltd, Tianjin, China) with a molar ratio Tb/C = 1:15, and the cathode was a graphite block (20 × 20 mm, Sinosteel Shanghai Carbon Plant Co., Ltd., Shanghai, China). Tb_4_O_7_ and graphite were the metal and carbon sources, respectively, for the formation of EMFs. An arc was generated under a He (Beijing AP Baif Gases Industry Co., Ltd., Beijing, China) atmosphere of 400 mbar at an electric current of 100 A. In comparison with the classical arc discharge method [39,40], a metal source of a higher oxidation state and a He atmosphere of a higher pressure were applied to the preparation of metallofullerenes, based on the optimization of the experimental conditions. 

When the arc chamber had cooled to room temperature, the yielded soot was collected and then extracted with 1,2,4-trichlorobenzene (Aladdin, Shanghai, China) in a round-bottomed flask connected to a condenser at about 387 K for 12 h. After removal of 1,2,4-trichlorobenzene using a rotary evaporator, the residue powder was immediately re-dissolved in the mixed solvent of CS_2_/toluene (Tianjin City Guang Fu Tech. Development Co., Ltd., Tianjing, China) (1:5, *v*/*v*) under the action of ultrasonic waves. In comparison with toluene, carbon disulfide (CS_2_, Aladdin, Shanghai, China) has better solubility toward EMFs and can be easily removed by rotary evaporation at room temperature. Thus, instead of pure toluene, the mixed solvent of CS_2_/toluene was applied to the redissolution of fullerenes and EMFs.

The isolation of Tb@C_82_ (I, II) isomers was conducted on a recycling preparative high-performance liquid chromatograph (HPLC, LC-9104, Analytical Co., Ltd., Kyoto, Japan) using toluene as the eluent. Three columns, including 5PYE (25 × 250 mm, NacalaiTesque, Kyoto, Japan), Buckyprep-M (25 × 250 mm, NacalaiTesque, Kyoto, Japan), and Buckyprep (25 × 250 mm, NacalaiTesque, Kyoto, Japan), were applied in a three-stage separation procedure. The purity of isolated Tb@C_82_ (I, II) isomers was checked by analytical HPLC (L-2000, Hitachi, Tokyo, Japan) and a matrix-assisted laser desorption/ionization time-of-flight (MALDI-TOF) mass spectrometer (Biflex III, Bruker, Bremen, Germany). The analytical HPLC was carried out on a Buckyprep column (4.6 × 250 mm, NacalaiTesque, Kyoto, Japan), and the matrix for MALDI mass spectrometry was 1,1,4,4-tetraphenyl-1,3-butadiene (TPB, Aladdin, Shanghai, China). 

The optical absorption spectra of the Tb@C_82_ (I, II) isomers were recorded by a UV750 spectrometer (Perkin Elmer LAMBDA, Waltham, MA, USA), with a CS_2_ solution of Tb@C_82_ (I) or Tb@C_82_ (II) in a quartz cell (1 cm). Electrochemical measurements were by a CHI-660E (Chenhua Energy Co., Ltd., Shanghai, China) instrument in a conventional three-electrode cell consisting of a platinum plate working electrode, a platinum plate as the counter-electrode, and one Ag electrode as a reference electrode. O-dichlorobenzene (o-DCB, Aladdin, Shanghai, China) was treated with magnesium strips to remove oxygen and water from the solution. The CV and DPV measurements were performed in o-dichlorobenzene (o-DCB) containing tetra(n-butyl) ammonium hexafluorophosphate ((n-Bu)_4_NPF_6_), Aladdin, Shanghai, China) (0.05 M), with a scan rate of 20 mV·s^−1^ at room temperature. All redox potentials were recorded on the Ag reference electrode and corrected according to Fc^0/+^.

### 2.2. Single-Crystal X-ray Diffraction Analysis

Ni(OEP) composed of C, Ni, H and N elements was applied as the cocrystal agent for the formation of cocrystals with Tb@C_82_ (I, II), in which Ni coordinates with porphyrin via four Ni-N coordinated bonds. The π-π interaction between Ni(OEP) (Aladdin, Shanghai, China) and Tb@C_82_ (I, II) makes it possible to grow cocrystals of Tb@C_82_ (I, II) and Ni(OEP). Benzene (Aladdin, Shanghai, China) and carbon disulfide are good solvents for Ni(OEP) and Tb@C_82_ (I, II), respectively, which could also crystallize with Ni(OEP) and Tb@C_82_ (I, II) to form cocrystals.

A slow solvent diffusion process was applied to the growth of cocrystals of Tb@C_82_ (I, II) isomers with Ni(OEP). A benzene solution of Ni(OEP) was put on a CS_2_ solution of Tb@C*_2v_*(9)-C_82_ or Tb@C*_S_*(6)-C_82_ in a 5.0 mL centrifuge tube. After standing still at 275 K for 3 weeks in a refrigerator, black cocrystals appeared at the interface of benzene and CS_2_, which were large enough for X-ray single-crystal diffraction analyses. The crystallographic data were collected on a diffractometer (D8 VENTURE, Bruker Analytik GmbH), and a multi-scan absorption correction was applied to the intensity data using the SADA program. For the cocrystal of Tb@C*_2v_*(9)-C_82_ with Ni(OEP), the data were collected by a CCD collector at 173 K under a radiation wavelength of 0.65250 Å. In contrast, for the cocrystal of Tb@C*_S_*(6)-C_82_ with Ni(OEP), the data were collected by a Photon100 CMOS collector at 100 K under a radiation wavelength of 0.88560 Å. The structures were solved using direct methods and refined by the SHELXL program. The crystallographic data for this paper can be found at CCDC1878018 and CCDC1878019, respectively. 

### 2.3. Theoretical Calculation Details

Density functional theory (DFT) calculations were carried out using the M06-2X [41] functional in conjunction with the 6-31G* all-electron basis sets for the C [42,43] and SDD [44] basis sets along with the corresponding relativistic small-core effective core potential for Tb (denoted as M06-2X/6-31G*~SDD). The DFT computations were calculated using the Gaussian 09 software package [45]. The results were visualized using the Mercury program [46].

## 3. Results and Discussion

### 3.1. Multistage HPLC Isolation of Tb@C_82_ (I, II) Isomers

Previously, pure endohedral metallofullerenes have been isolated by preparative high-performance liquid separation chromatography (HPLC) without exception, in which various columns are applied in a multi-stage procedure. In this work, Tb@C_82_ (I, II) isomers were isolated by the following process. The toluene solution of fullerenes and metallofullerenes was filtered with a 0.2 μm PTFE membrane, then subjected to three-stage HPLC isolation. Firstly, a 5PYE column was chosen to separate the mixture of fullerenes and metallofullerenes. The 5PYE was packed with planar aromatic 2-(1-pyrene)ethyl-group-modified silica gel, which is appropriate to separate fullerenes and metallofullerenes in line with their molecular weights. As shown in Figure 1a,b, the eluate collected at 32–39.5 min was mainly composed of C_88_ and Tb@C_82_ (I, II). Secondly, a Buckyprep-M column was adopted to separate C_88_ from Tb@C_82_ (I, II). The Buckyprep-M column was packed with phenothiazine-bonded silica gel, which plays a crucial role in isolating metallofullerenes from fullerenes. It can be seen in Figure 1c,d that the eluate collected at 33.5–40 min was mainly composed of Tb@C_82_ (I, II). Finally, the Tb@C_82_ (I, II) isomers were isolated on a Buckyprep column by cyclic HPLC. The Buckyprep column was packed with 3-(1-pyrene)propyl-bonded silica gel, which has good ability to isolate metallofullerene isomers. As displayed in Figure 1e, Tb@C_82_ (I) and Tb@C_82_ (II) were successfully isolated after five cycles.

### 3.2. Purity Evaluation and Spectroscopic Characterization

The purity of the isolated Tb@C_82_ (I, II) samples was evaluated by analytical HPLC chromatograms and MALDI-TOF mass spectra. It can be seen in Figure 2a that the retention times of the Tb@C_82_ (I, II) isomers on the Buckyprep column are quite different, and the two isomers demonstrate a single HPLC peak with quite good symmetry, respectively. Shown in Figure 2b are the MALDI-TOF spectra of Tb@C_82_ (I, II), which demonstrate a single peak at *m*/*z* 1143, corresponding to the molecular mass of Tb@C_82_. Moreover, the observed isotopic distributions agree very well with the corresponding theoretically calculated ones. Therefore, it is reasonable for us to conclude that the purity of the isolated Tb@C_82_ (I, II) isomers is more than 95%.

The vis-NIR absorption spectra of Tb@C_82_ (I, II) are shown in Figure 3. Tb@C_82_ (I) shows three distinct peaks at 635, 998, and 1413 nm, and two characteristic peaks at 780 and 1089 nm are identified for Tb@C_82_ (II). The visible and NIR absorption features of EMFs are owing to the π-π* transitions of the fullerene cages, which are directly relate to the symmetry and charge state of the carbon cages. Therefore, information on the symmetry of the carbon cages could be extracted from the optical absorption spectroscopy of EMFs. The vis-NIR absorption spectra of Tb@C*_2v_*(9)-C_82_ and Tb@C*_S_*(6)-C_82_ are substantially similar to these of Y@C*_2v_*(9)-C_82_ and Y@C*_S_*(6)-C_82_, whose molecular structures have been determined by X-ray single-crystal diffraction [36]. Accordingly, the carbon cages of Tb@C_82_ (I) and Tb@C_82_ (II) are tentatively ascribed to C*_2v_* and C*_S_* symmetries, and the oxidation state of the endohedral Tb atom might be +3.

### 3.3. Crystallographic Study

In line with the crystallographic data and structural refinement statistics of the cocrystals obtained in this work (Appendix A), Tb@C_82_ (I) crystallizes with Ni(OEP) in the common *C2*/*m* space group, and a pair of disordered C*_2v_*-C_82_ cages can be identified. Among them, half of the Ni(OEP) molecule and both halves of the C_82_ cage are located on one side of the lens plane in the asymmetric unit, and the equal phase occupancies are 0.5. Tb@C*_S_*(6)-C_82_ isomers crystallized in the commonly encountered space group P2_1/c_. Both cocrystals belong to a monoclinic crystal system, and based their crystallographic mirror planes, are not consistent with any symmetrical elements of the C_82_ cages.

The molecular structures of the two Tb@C_82_ isomers were unambiguously determined by means of single-crystal XRD. As shown in Figure 4a, the fullerene cage of Tb@C_82_ (I) is clearly ascribed to C*_2v_*(9)-C_82_. Inside the cage, the Tb atom shows three disordered positions with fractional occupancy values of 0.6722 (Tb1), 0.2222 (Tb2), and 0.1056 (Tb3), respectively, indicative of a motional behavior (see Appendix A). Moreover, all of these metal positions are off the center of the carbon cage to some extent. The major metal atom (Tb1) resides near a [6,6]-bond (C47-C48), and the shortest terbium–cage distances are 2.313 Å for Tb to C47 and 2.308 Å for Tb to C48 (see Figure 5a). Contrastingly, the fullerene cage of Tb@C_82_ (II) is definitely assigned to C*_S_*(6)-C_82_, in which four disordered positions of the embedded Tb atom are identified with fractional occupancy values of 0.1069 (Tb1), 0.7951 (Tb2), 0.0499 (Tb3), and 0.0484 (Tb4), respectively (see Appendix A), in sharp contrast to the fixed positions of Y^3+^ and Er^3+^ inside C*_S_*(6)-C_82_ observed in their corresponding cocrystals with Ni(OEP) [36,37]. The major Tb (Tb2) resides opposite a pentagon−hexagon−hexagon junction, and the shortest metal–cage distance is 2.288 Å for Tb2 to C65 (see Figure 5b), which is a little bit shorter than that observed in C*_2v_*(9)-C_82_. Moreover, it can be seen from Figure 4b that two Tb@C*_S_*(6)-C_82_ molecules are linked via a C−C bond whose bond length is 1.624 Å, and a dimeric structure is formed similar to the dimers of Y@C*_S_*(6)-C_82_ and Er@C*_S_*(6)-C_82_ discovered in their corresponding cocrystals with Ni(OEP). In addition, the Tb@C*_S_*(6)-C_82_ molecules adopt an orientation in the dimer to keep the encaged Tb atoms as far away as possible.

Figure 6 depicts the packing structures of the two cocrystals of isomeric Tb@C_82_ (I, II) and Ni(OEP). It can be seen in Figure 6a that monomeric Tb@C*_2v_*(9)-C_82_ molecules form zigzag chains in this polymorph, while Ni(OEP) molecules are aligned in two dimensions, forming a closely packed bilayer with a gap of 3.1122 Å. The Tb@C*_2v_*(9)-C_82_ molecule chains pack closely between the Ni(OEP) bilayers, which are placed 20.7846 Å apart. In contrast, as indicated in Figure 6b, the Ni(OEP) bilayers placed 23.7195 Å apart do not align in two-dimensional planes and dimerization occurs for Tb@C*_S_*(6)-C_82_ molecules in the cocrystal of Tb@C*_S_*(6)-C_82_ and Ni(OEP), similar to the arrangement of Y@C*_S_*(6)-C_82_ and Er@C*_S_*(6)-C_82_ in their corresponding cocrystals with Ni(OEP) [36,37]. Of interest to note is that the gap in the Ni(OEP) bilayer (3.184 Å) is larger but the distance between Ni(OEP) and the carbon cage (2.861 Å) is shorter than those observed in Figure 6a. Thus, a weaker π-π interaction in the Ni(OEP) bilayers and a stronger π-π interaction between Ni(OEP) and the carbon cage are expected in the cocrystal of Tb@C*_S_*(6)-C_82_ and Ni(OEP) [47].

### 3.4. Computational Study

DFT calculations at the M06-2X/6-31G*~SDD level of theory were then carried out to understand the different packing behaviors of the two Tb@C_82_ isomers. According to the electronic configuration of element Tb[Xe]4f^9^6s^2^ and its formal charge of 3+, different spin multiplicities were considered for each Tb@C_82_ isomer during the geometric optimization. Appendix A shows that Tb@C*_S_*(6)-C_82_ and Tb@C*_2v_*(9)-C_82_ both have open-shell electronic structures due to the presence of odd electrons, but they energetically prefer the sextet and octet electronic states as their ground states, respectively.

Electrons could be spin-up and -down while distributing on an orbital, which are marked as α and β spin states, respectively. Spin density reflects the difference in the density distributions of α and β spin electrons in an open-shell system. In contrast, the spin density is zero for a closed-shell system at ground state because the α and β spin electrons are equal. The quantitative calculations give the electron spin density of a molecule. The larger the absolute value of spin density is, the more single electrons there are after spatial integration, the more chances there are for the whole cluster system to lose or gain electrons, and the more active the system. The electron spin density obtained by calculation helps to interpret the experimental data. With accurate theory, calculated electron spin density can even verify or predict experimental results.

To rationalize the experimental results, the spin-density distributions of Tb@C*_S_*(6)-C_82_ and Tb@C*_2v_*(9)-C_82_ were then calculated and compared (Figure 7). The spin density was the largest (ca. 6 a.u.) at the internal Tb^3+^ cation due to its 4f^8^ electronic configuration (six unpaired electrons according to Hund’s rule), since the Tb atom donates one 4f and two 6s electrons to the outer C_82_ cage. After accepting the three electrons from Tb, the C_82_^3−^ cage has an unpaired electron and thus also has an obvious spin-density distribution. Remarkably, C44 on the Tb@C*_S_*(6)-C_82_ surface exhibits the largest spin-density value (−0.22 a.u.) among the cage carbon atoms, indicating that it is an ideal site for the exohedral dimerization reaction to achieve spin pairing between two monomers. Indeed, our experiment shows that exactly this reaction site is involved during the dimer formation. A similar situation exists in Y@C*_S_*(6)-C_82_ and Er@C*_S_*(6)-C_82_, in which the carbon atoms at the site of dimerization also possess the highest spin-density value [36,37]. As for Tb@C*_2v_*(9)-C_82_, however, all the carbon atoms have moderate spin density, suggesting their low reactivity. Therefore, the different dimerization behaviors between Tb@C*_S_*(6)-C_82_ and Tb@C*_2v_*(9)-C_82_ under the same crystallization conditions should be attributed to their rather different spin-density distributions on their C_82_ cages.

### 3.5. Electrochemical Study 

The redox properties of Tb@C*_2v_*(9)-C_82_ and Tb@C*_S_*(6)-C_82_ were investigated by cyclic voltammetry (CV) and differential pulse voltammetry (DPV, see Appendix A). It can be seen in Figure 8 that both Tb@C*_2v_*(9)-C_82_ and Tb@C*_S_*(6)-C_82_ display two oxidation and five reduction DPV peaks, whose corresponding potentials are listed in Table 1. In comparison with those reported for M@C_82_ (I, II) (M = Y [36], Er [37], and Nd [48]), the redox potentials obtained here for Tb@C_82_ (I, II) largely enrich the electrochemistry of C_82_-based mono-EMFs. Since both the first oxidation and first reduction potentials of Tb@C*_S_*(6)-C_82_ are lower than those of Tb@C*_2v_*(9)-C_82_, respectively, it is concluded that the former is accordingly more easily oxidized or reduced than the latter. Moreover, the electrochemical band gaps (∆E = E_1/2_, ox^(1)^-E_1/2_, _red_^(1)^) of Tb@C*_2v_*(9)-C_82_ and Tb@C*_S_*(6)-C_82_ are calculated to be 0.49 and 0.37 V, respectively, confirming that the thermodynamic stability of Tb@C*_2v_*(9)-C_82_ is higher than that of Tb@C*_S_*(6)-C_82_.

## 4. Conclusions

In summary, two Tb@C_82_ (I, II) isomers were isolated by multi-stage preparative high-performance liquid chromatography (HPLC). Optical absorption spectra indicate that the carbon cages of Tb@C_82_ (I, II) are of C*_2v_* and C*_S_* symmetry, respectively, and the formal oxidation state of endohedral Tb cation is +3. Electrochemical studies evidence that the thermodynamic stability of Tb@C*_2v_*(9)-C_82_ is higher than that of Tb@C*_S_*(6)-C_82_. With the successful growth of cocrystals of Tb@C_82_ (I, II) with Ni(OEP), the various C*_2v_* and C*_S_* symmetries of carbon cages as well as the disordered sites of endohedral Tb cations were explicitly determined by crystallographic study. Moreover, in their corresponding cocrystals with Ni(OEP), Tb@C*_S_*(6)-C_82_ molecules form dimers via a covalent C-C bond, while monomeric Tb@C*_2v_*(9)-C_82_ molecules assemble into a zig-zag chain. DFT calculations suggest that a specific cage carbon atom with an abnormally high spin density is responsible for the regioselective dimerization of Tb@C*_S_*(6)-C_82_ molecules.

## Figures and Tables

**Figure 1 nanomaterials-13-00994-f001:**
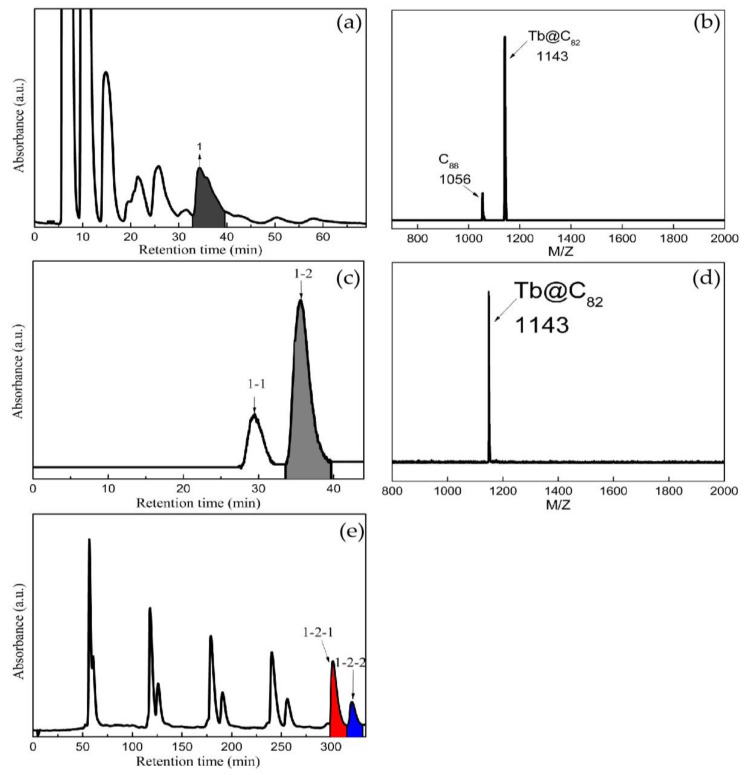
The (**a**) first, (**c**) second and (**e**) third stages of HPLC isolation of Tb@C_82_ (I, II) isomers (column size: Ф20 mm × 250 mm; injection volume: 20 mL; eluent: toluene with a flow rate of 10 mL/min; detection wavelength: 290 nm at 298 K). The MALDI-TOF mass spectra of (**b**) fraction 1 and (**d**) fraction 1-2.

**Figure 2 nanomaterials-13-00994-f002:**
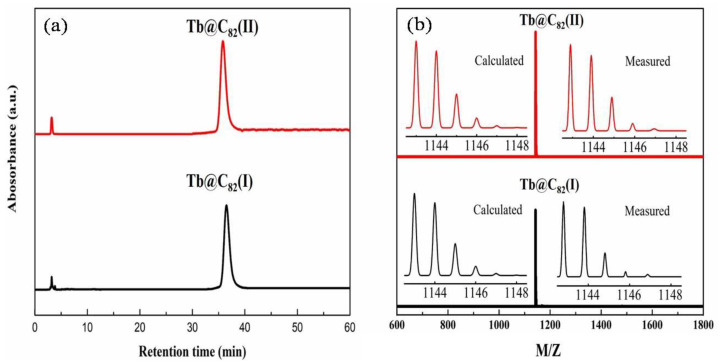
(**a**) Analytical liquid chromatography chromatograms on a Buckyprep column (Φ = 4.6 × 250 mm, 1 mL/min in toluene flow) and (**b**) MALDI-TOF mass spectra of Tb@C_82_ (I, II) isomers.

**Figure 3 nanomaterials-13-00994-f003:**
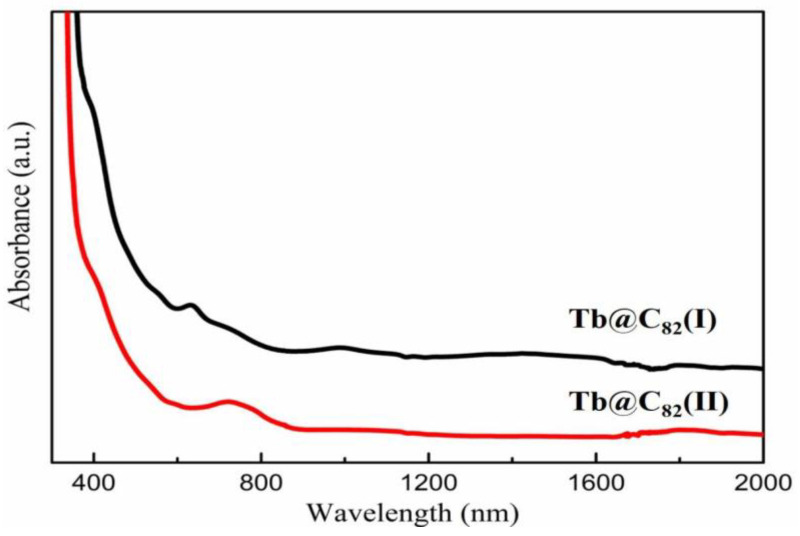
The vis-NIR absorption spectra of Tb@C_82_ (I, II) isomers in CS_2_ at 298 K.

**Figure 4 nanomaterials-13-00994-f004:**
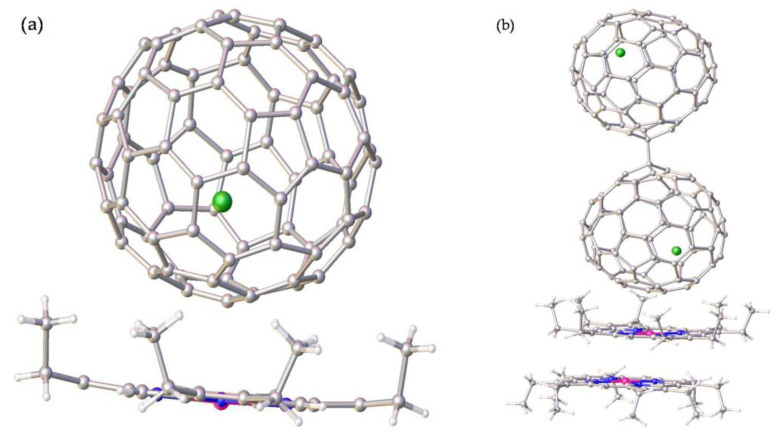
The orientations of (**a**) Tb@C*_2v_*(9)-C_82_ and (**b**) Tb@C*_S_*(6)-C_82_ to Ni(OEP) in the cocrystals of Tb@C*_2v_*(9)-C_82_]∙[Ni(OEP)] and 2[Tb@C*_S_*(6)-C_82_]∙2[Ni(OEP)], respectively. Only the major Tb site and one carbon cage orientation are shown. Solvent molecules and H atoms are omitted for clarity.

**Figure 5 nanomaterials-13-00994-f005:**
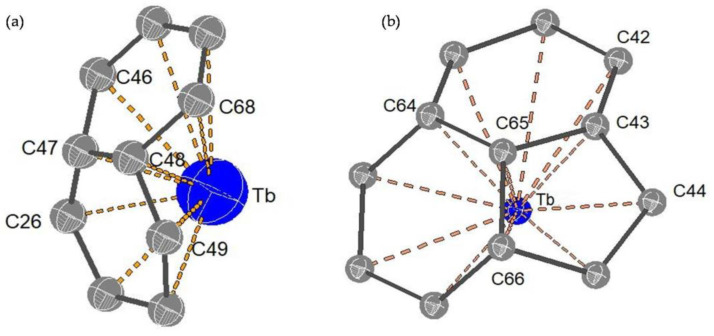
The interaction of the metal cation (major metal site) with the nearest cage carbon atoms for (**a**) Tb@C*_2v_*(9)-C_82_ and (**b**) Tb@C*_S_*(6)-C_82_.

**Figure 6 nanomaterials-13-00994-f006:**
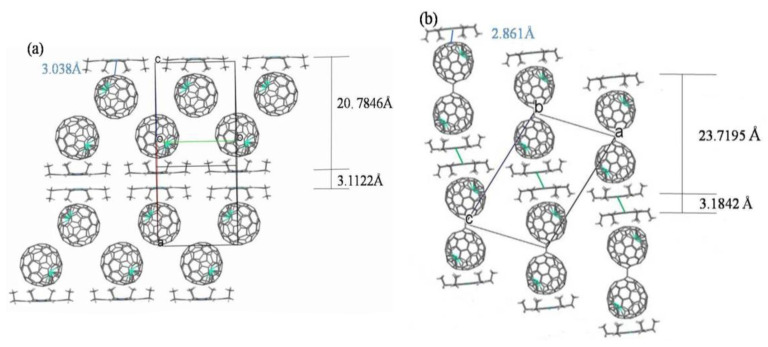
The packing structures of (**a**) Tb@C*_2v_*(9)-C_82_ and (**b**) Tb@C*_S_*(6)-C_82_ in their cocrystals with Ni(OEP). Only one cage orientation and the major terbium site are illustrated, with solvent molecules omitted for clarity.

**Figure 7 nanomaterials-13-00994-f007:**
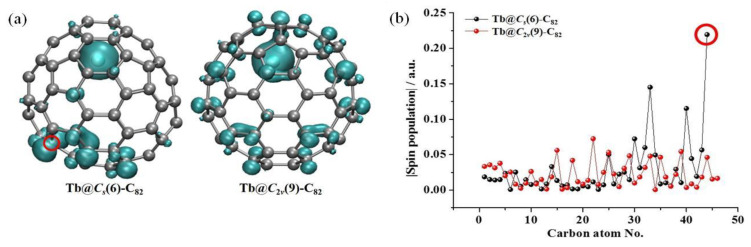
(**a**) Visualized spin density distributions and (**b**) corresponding spin population values of unequivalent cage carbon atoms of Tb@C*_S_*(6)-C_82_ and Tb@C*_2v_*(9)-C_82_. The absolute values are used for easy comparison. Please refer to Appendix A for the numbering schemes. The C44 site of Tb@C*_S_*(6)-C_82_, with the largest spin density among all cage carbons, is highlighted by the red circle.

**Figure 8 nanomaterials-13-00994-f008:**
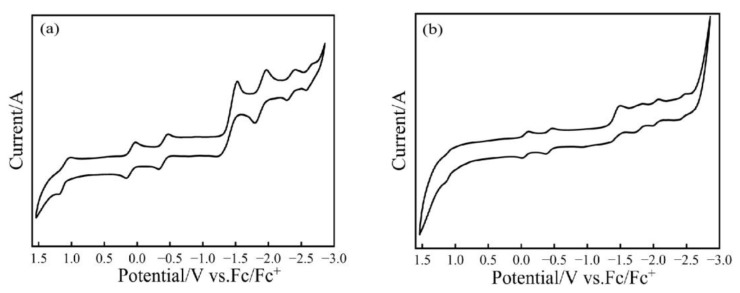
CV curves of (**a**) Tb@C*_2v_*(9)-C_82_ and (**b**) Tb@C*_S_*(6)-C_82_ isomers. Conditions: working electrode, platinum plate; counter electrode, platinum plate; reference electrode, Ag wire; supporting electrolyte, 0.05 M n-Bu_4_NPF_6_ in o-dichlorobenzene; scan rate, 20 mV/s.

**Table 1 nanomaterials-13-00994-t001:** Redox Potentials (V vs. Fc/Fc^+^) ^a^ of Tb@C*_2v_*(9)-C_82_ and Tb@C*_S_*(6)-C_82_.

EMFs	^ox^*E*_2_ (V)	^ox^*E*_1_ (V)	^red^*E*_1_ (V)	^red^*E*_2_ (V)	^red^*E*_3_ (V)	^red^*E*_4_ (V)	^red^*E*_5_ (V)	EC Gap (V)
Tb@C*_2v_*(9)-C_82_	1.10	0.09	−0.40	−1.36	−1.87	−2.34	−2.61	0.49
Tb@C*_S_*(6)-C_82_	1.10	−0.05	−0.42	−1.43	−1.76	−2.04	−2.43	0.37

^a^ Half-wave potentials in o-DCB unless otherwise addressed.

## Data Availability

CCDC1878018 and CCDC1878019 contain the supplementary crystallographic data for this paper. These data can be obtained free of charge via https://www.ccdc.cam.ac.uk/structures/ (accessed on 3 February 2022), or by emailing data_request@ccdc.cam.ac.uk, or by contacting The Cambridge Crystallographic Data Centre, 12 Union Road, Cambridge CB2 1EZ, UK; fax: +44 1223336033.

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
