# Peer review of "The Various Packing Structures of Tb@C82 (I, II) Isomers in Their Cocrystals with Ni(OEP)"

_nanomaterials, 2023, doi:10.3390/nano13060994_

Round 1

Reviewer 1 Report

This manuscript provides a comprehensive overview of the various packing structures of Tb@C82 (I, II) isomers in their cocrystals with Ni(OEP). The authors first use multistage HPLC isolation to separate the isomers, followed by spectroscopic and crystallographic characterization. The authors then perform density functional theory calculations to explain the regional selectivity of the dimerization of Tb@C82(II). Lastly, the authors present electrochemical studies which indicate the thermodynamic stability of Tb@C2v(9)-C82 is higher than that of Tb@C82(II). Overall, the authors have presented a thorough investigation of the packing structures of Tb@C82 isomers in their cocrystals with Ni(OEP). The manuscript is clear and the results are robust. Therefore, the manuscript can be accepted with minor revisions.

Please respond to the following suggestions and provide more detail in the manuscript:

1. Add more details on the preparation, isolation, characterization and studies of the Tb@C82 isomers.
2. Include more insights on the role of the various elements involved in the studies.
3. Explain in detail the role of the HPLC in the isolation of Tb@C82 isomers.
4. Incorporate additional citations from other research on the use of fullerenes in relation to the topic. For example DOI: 10.1038/s41557-021-00658-6 DOI: 10.1016/j.molliq.2018.08.069 DOI: 10.1002/adfm.202104369
DOI: 10.1016/j.dyepig.2020.108918 DOI: 10.1002/adma.202004115
5. Explain the spin-density distribution in detail.

This manuscript needs to have its scientific writing improved.
1. "rotary evaporation, the mixture was dissolved in toluene by ultrasonication" - "rotary evaporation, the mixture was dissolved into toluene by ultrasonication"
2. "which was collected Tb@C82 (I, II) isomers" - "which collected Tb@C82 (I, II) isomers"
3. "It is clearly characterized by X-ray single crystal diffraction Tb@C82 (I, 16 II)" - "It is clearly characterized by X-ray single crystal diffraction of Tb@C82 (I, II)"
4. "the molecular structure of (I, II) is Tb@C2v(9)-C82 and Tb@Cs(6)-C82, respectively" - "the molecular structure of (I, II) are Tb@C2v(9)-C82 and Tb@Cs(6)-C82, respectively"
5. "the crude production in toluene" - "the crude product in toluene"
6. "the spectra of Tb@C2v(9)-C82 and Tb@Cs(6)-C82 are substantially similar" - "the spectra of Tb@C2v(9)-C82 and Tb@Cs(6)-C82 are substantially similar to"
I noticed several typos and grammar mistakes in my quick review. It is the authors' responsibility to ensure that the scientific writing in the entire manuscript is improved.

Reviewer 2 Report

The manuscript provided by W. Dong et al. describes the isolation of two isomers of Tb@C82 and the preparation of their cocrystals with Ni(OEP). First of all, the Authors should carefully revise this work, and correct typos appearing within the text (see, for example, lines 26, 290, 291). On the first page, the ref. 3 is missing. I do not understand the sentence: "The crude ....film", line 133, line 170 - the same. Also, the formatting of many references leaves much to be desired. Besides these I have some concerns about the scientific aspect of this work:

1. The Authors state that the soot containing Tb@C82 isomers was synthesized by an improved discharge method. What is this improvement? Relevant references to the classical method are needed.

2. Figure 6b is of low quality and should be improved.

3. The section devoted to the DFT theoretical calculations is too lapidary.

To conclude the reviewed manuscript probably can be published in the Nanomaterials Journal after major revision and addressing the above-listed points.

Round 2

Reviewer 2 Report

The corrections made by the Authors are satisfactory. The manuscript can be published in the Nanomaterials Journal in its present form.